# Einstein-AdS Gravity Coupled to Nonlinear Electrodynamics, Magnetic Black Holes, Thermodynamics in an Extended Phase Space and Joule–Thomson Expansion

**Sergey Il'ich Kruglov** [1,2]

1   Department of Physics, University of Toronto, 60 St. Georges St., Toronto, ON M5S 1A7, Canada;
    serguei.krouglov@utoronto.ca
2   Canadian Quantum Research Center, 204-3002 32 Ave, Vernon, BC V1T 2L7, Canada

**Abstract:** We studied Einstein's gravity with negative cosmological constant coupled to nonlinear electrodynamics proposed earlier. The metric and mass functions and corrections to the Reissner–Nordström solution are obtained. Black hole solutions can have one or two horizons. Thermodynamics and phase transitions of magnetically charged black holes in Anti-de Sitter spacetime are investigated. The first law of black hole thermodynamics is formulated and the generalized Smarr relation is proofed. By calculating the Gibbs free energy and heat capacity we study the black hole stability. Zero-order (reentrant), first-order, and second-order phase transitions are analyzed. The Joule–Thomson expansion is considered, showing the cooling and heating phase transitions. It was shown that the principles of causality and unitarity are satisfied in the model under consideration.

**Keywords:** Einstein's gravity; nonlinear electrodynamics; black hole thermodynamics; phase transitions

## 1. Introduction

Black holes behave as thermodynamic systems [1–3] and they have the entropy connected with the surface area. The surface gravity defines the temperature [4,5]. Black holes phase transitions occur in Anti-de Sitter (AdS) spacetime, where the cosmological constant is negative [6]. It was discovered that gravity in AdS spacetime is linked with the conformal field theory (the holographic principle) [7] that has an application in condensed matter physics. In black hole thermodynamics (in an extended phase space), the negative cosmological constant is a thermodynamic pressure which is conjugated to a black hole volume [8–11]. In Einstein-AdS gravity coupled to nonlinear electrodynamics (NED) with coupling $\beta$, the constant $\beta$ is conjugated to the vacuum polarization. The first NED was proposed by M. Born and L. Infeld [12] to remove a singularity of a point charge and to obtain the finite the electromagnetic field energy. At the weak-field limit, Born–Infeld electrodynamics becomes Maxwell's theory. Another NED model was formulated by W. Heisenberg and H. Euler [13], where nonlinearity is due to the creation of the electron-positron pairs within quantum electrodynamics. The interest in NED, as a source of gravity is because of the possibility of having regular black holes and soliton-like configurations without singularities. Recent reviews of NED models were given in [14,15]. Black hole thermodynamics in Einstein-AdS gravity coupled to Born–Infeld electrodynamics was considered in [16–22] (see also [23,24]). Born–Infeld-AdS thermodynamics of black holes in an extended phase space was studied in [25–29]. The Joule–Thomson expansion of black holes was investigated in [30–39]. In this paper we studied a modified Einstein-AdS theory with a NED model, as a matter field, to smooth out singularities of the linear Maxwell theory. We used NED theory with Lagrangian of the form $\mathcal{L}(\mathcal{F}) = -\mathcal{F}/\left(4\pi\left(1 + (2\beta\mathcal{F})^{3/4}\right)\right)$, where $\mathcal{F} = F^{\mu\nu}F_{\mu\nu}/4$, with $F_{\mu\nu}$ being the electromagnetic field tensor. The interest in this model is due to its simplicity—the metric and mass functions are expressed in the form of

elementary functions but in Born–Infeld NED they are special functions. We consider magnetically charged black holes because electrically charged black holes with NED possessing a weak-field Maxwell limit have a singularity [40]. It is worth mentioning that Lagrangians of NED models in the weak-field limit are different. This leads to different indexes of diffraction and birefringent effects. The similarities in the behavior of critical isotherms, the magnetic potential, vacuum polarization, the Gibbs free energy, and heat capacity take place for Einstein-AdS gravity coupled to NED models. Here, attention is paid to gravity in the AdS (not in de Sitter) spacetime because this case allows us to introduce a pressure which is necessary to consider an extended phase space and thermodynamics. In addition, the holographic principle only occurs in this case.

In Section 2 we obtain the metric function and its asymptotic with corrections to the Reissner–Nordström solution. The first law of black hole thermodynamics in the extended phase space is studied in Section 3. We calculate the thermodynamic magnetic potential and the thermodynamic conjugate to the NED coupling (the vacuum polarization). We show that the generalized Smarr relation holds. In Section 4, the critical temperature and critical pressure are obtained. By analysing the Gibbs free energy and heat capacity we show that phase transitions take place. It is demonstrated that black hole thermodynamics is similar to Van der Waals thermodynamics. We analyse first-order, second-order, and reentrant phase transitions. The Joule–Thomson adiabatic expansion is studied in Section 5. The Joule–Thomson coefficient and the inversion temperature are calculated. Section 6 is a summary. In Appendix A we calculate the Kretschmann scalar. We study causality and unitarity of our NED model in Appendix B. We show that the principles of causality and unitarity take place for any magnetic induction fields.

We use the units: $c = \hbar = 1, k_B = 1$.

## 2. Einstein-AdS Black Hole Solution

The Einstein-AdS action with the matter is given by

$$I = \int d^4x \sqrt{-g} \left( \frac{R - 2\Lambda}{16\pi G_N} + \mathcal{L}(\mathcal{F}) \right),$$
(1)

where $\Lambda = -3/l^2$ is negative cosmological constant, with $l$ being the AdS radius. Here, we use the matter Lagrangian in the form of NED[1] [41]

$$\mathcal{L}(\mathcal{F}) = -\frac{\mathcal{F}}{4\pi\left(1 + (2\beta\mathcal{F})^{3/4}\right)},$$
(2)

with $\mathcal{F} = F^{\mu\nu}F_{\mu\nu}/4 = (B^2 - E^2)/2$, where $E$ and $B$ are the electric and magnetic induction fields, respectively. As $\beta \to 0$ Lagrangian (2) becomes the Maxwell Lagrangian $\mathcal{L}_M = -\mathcal{F}/(4\pi)$. From action (1), one obtains the field equations

$$R_{\mu\nu} - \frac{1}{2}g_{\mu\nu}R + \Lambda g_{\mu\nu} = 8\pi G_N T_{\mu\nu},$$
(3)

$$\partial_\mu\left(\sqrt{-g}\mathcal{L}_\mathcal{F}F^{\mu\nu}\right) = 0,$$
(4)

where $\mathcal{L}_\mathcal{F} = \partial\mathcal{L}(\mathcal{F})/\partial\mathcal{F}$. The energy–momentum tensor reads

$$T_{\mu\nu} = F_{\mu\rho}F_\nu{}^\rho\mathcal{L}_\mathcal{F} + g_{\mu\nu}\mathcal{L}(\mathcal{F}).$$
(5)

The line element squared with spherical symmetry is

$$ds^2 = -f(r)dt^2 + \frac{1}{f(r)}dr^2 + r^2\left(d\theta^2 + \sin^2(\theta)d\phi^2\right).$$
(6)

We treat the black hole as a magnetic monopole with the magnetic induction field $B = q/r^2$, where $q$ is the magnetic charge. The metric function is given by [40]

$$f(r) = 1 - \frac{2m(r)G_N}{r}, \tag{7}$$

with the mass function

$$m(r) = m_0 + 4\pi \int_0^r \rho(r)r^2 dr. \tag{8}$$

Here, $m_0$ is an integration constant (the Schwarzschild mass) and $\rho$ is the energy density. Making use of Equation (5), the magnetic energy density plus the energy density due to AdS spacetime is given by

$$\rho = \frac{q^2}{8\pi r \left(r^3 + (\beta q^2)^{3/4}\right)} - \frac{3}{8\pi G_N l^2}. \tag{9}$$

From Equations (8) and (9) one obtains the mass function

$$m(r) = m_0 + \frac{q^{3/2}}{12\sqrt[4]{\beta}} \left[ \ln \frac{r^2 - \sqrt[4]{\beta q^2}\, r + \sqrt{\beta}\, q}{(r + \sqrt[4]{\beta q^2})^2} \right.$$
$$\left. -2\sqrt{3} \arctan \left( \frac{1 - 2r/\sqrt[4]{\beta q^2}}{\sqrt{3}} \right) + \frac{\pi}{\sqrt{3}} \right] - \frac{r^3}{2G_N l^2}. \tag{10}$$

The magnetic energy of the black hole becomes

$$m_M = \frac{q^2}{2} \int_0^\infty \frac{r}{r^3 + (\beta q^2)^{3/4}} dr = \frac{\pi q^{3/2}}{3\sqrt{3}\sqrt[4]{\beta}}. \tag{11}$$

The magnetic energy, which can be considered as a magnetic mass, is finite. Thus, the coupling $\beta$ smoothes singularities. Making use of Equations (7) and (10), we obtained the metric function

$$f(r) = 1 - \frac{2m_0 G_N}{r} - \frac{q^{3/2}G_N}{6\sqrt[4]{\beta}r} \left[ \ln \frac{r^2 - \sqrt[4]{\beta q^2}\, r + \sqrt{\beta}\, q}{(r + \sqrt[4]{\beta q^2})^2} \right.$$
$$\left. -2\sqrt{3} \arctan \left( \frac{1 - 2r/\sqrt[4]{\beta q^2}}{\sqrt{3}} \right) + \frac{\pi}{\sqrt{3}} \right] + \frac{r^2}{l^2}. \tag{12}$$

As $r \to 0$, when the Schwarzschild mass is zero ($m_0 = 0$), one finds

$$f(r) = 1 - \frac{G_N \sqrt{q} r}{2\beta^{3/4}} + \frac{r^2}{l^2} + \frac{G_N r^4}{5\beta^{3/2} q} + \mathcal{O}(r^6). \tag{13}$$

As a result, we have $f(0) = 1$. The finiteness of the metric function is necessary condition in order to have the spacetime regular. But the spacetime singularity is present in the model because the Kretschmann scalar becomes infinite at $r = 0$ (see Appendix A). Making use of Equation (12) (when $\Lambda = 0$) as $r \to \infty$, we obtain

$$f(r) = 1 - \frac{2MG_N}{r} + \frac{q^2 G_N}{r^2} - \frac{q^{7/2}\beta^{3/4}G_N}{4r^5} + \mathcal{O}(r^{-6}). \tag{14}$$

We define $M = m_0 + m_M$ being the ADM mass. According to Equation (14), black holes have corrections to the Reissner–Nordström solution. When $\beta = 0$, the metric (14) becomes the Reissner–Nordström metric. The plot of metric function (12) is depicted in Figure 1 (at $m_0 = 0$, $G_N = 1$, $l = 10$).

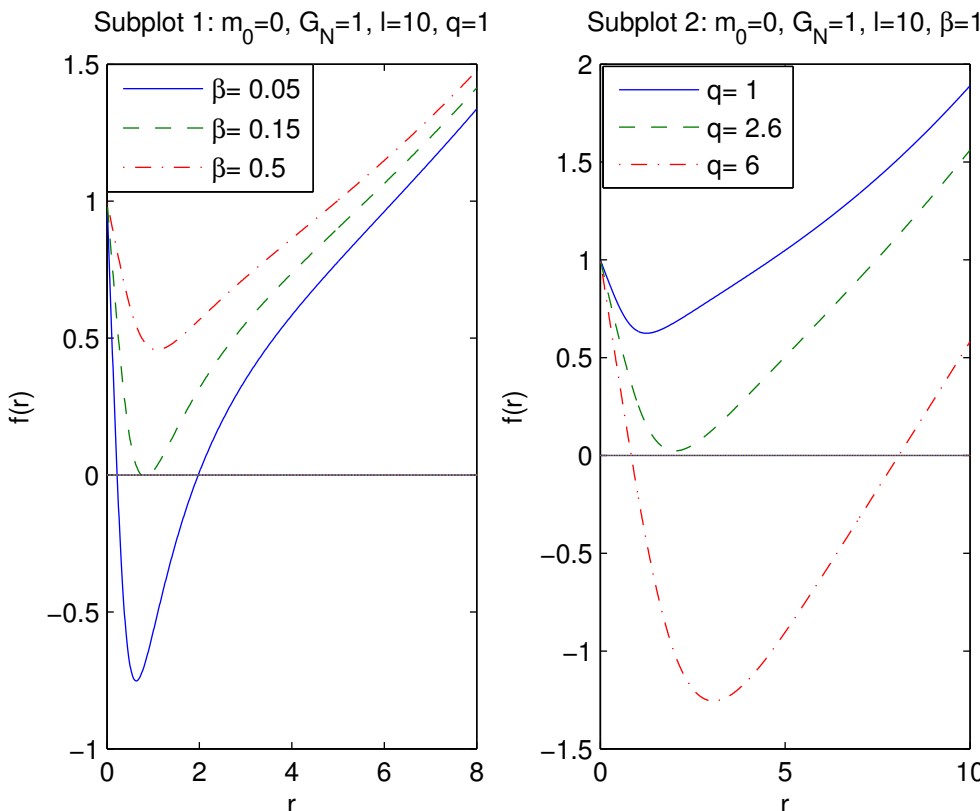

**Figure 1.** The function $f(r)$ at $m_0 = 0$, $G_N = 1$, $l = 10$. Figure 1 shows that black holes could have one or two horizons. In accordance with subplot 1, if coupling $\beta$ is increasing, the event horizon radius decreases. According to subplot 2, when magnetic charge $q$ increases the event horizon radius also increases.

According to Figure 1, black holes may have one or two horizons. When coupling $\beta$ increases, at the constant $q$, the event horizon radius is decreasing. If magnetic charge $q$ increases, at the constant $\beta$, the event horizon radius also increases.

### 3. First Law of Black Hole Thermodynamics

We will consider the first law of black hole thermodynamics in extended phase space, where the pressure is $P = -\Lambda/(8\pi)$ [42–46] and coupling $\beta$ is the thermodynamic value. In this approach, mass $M$ is a chemical enthalpy ($M = U + PV$ with $U$ being the internal energy). By using the Euler's dimensional analysis with $G_N = 1$ [42,47], we obtain dimensions as follows: $[M] = L$, $[S] = L^2$, $[P] = L^{-2}$, $[J] = L^2$, $[q] = L$, $[\beta] = L^2$. Then one finds

$$M = 2S\frac{\partial M}{\partial S} - 2P\frac{\partial M}{\partial P} + 2J\frac{\partial M}{\partial J} + q\frac{\partial M}{\partial q} + 2\beta\frac{\partial M}{\partial \beta}, \tag{15}$$

where $J$ is the black hole angular momentum. The thermodynamic conjugate to coupling $\beta$ is the vacuum polarization [11] $\mathcal{B} = \partial M/\partial \beta$. The black hole entropy $S$, volume $V$, and pressure $P$ are defined as

$$S = \pi r_+^2, \quad V = \frac{4}{3}\pi r_+^3, \quad P = -\frac{\Lambda}{8\pi} = \frac{3}{8\pi l^2}. \tag{16}$$

Making use of Equation (12) for non-rotating black holes ($J = 0$), we obtain

$$M(r_+) = \frac{r_+}{2G_N} + \frac{r_+^3}{2G_N l^2} + \frac{\pi q^{3/2}}{4\sqrt{3}\sqrt[4]{\beta}} - \frac{q^{3/2}g(r_+)}{12\sqrt[4]{\beta}},$$

$$g(r_+) = \ln \frac{r_+^2 - \sqrt[4]{\beta q^2} r_+ + \sqrt{\beta} q}{(r_+ + \sqrt[4]{\beta q^2})^2} - 2\sqrt{3} \arctan\left(\frac{1 - 2r_+ / \sqrt[4]{\beta q^2}}{\sqrt{3}}\right), \tag{17}$$

where $r_+$ is the event horizon radius, $f(r_+) = 0$. The total black hole mass $M(r_+)$ versus $r_+$ is plotted in Figure 2.

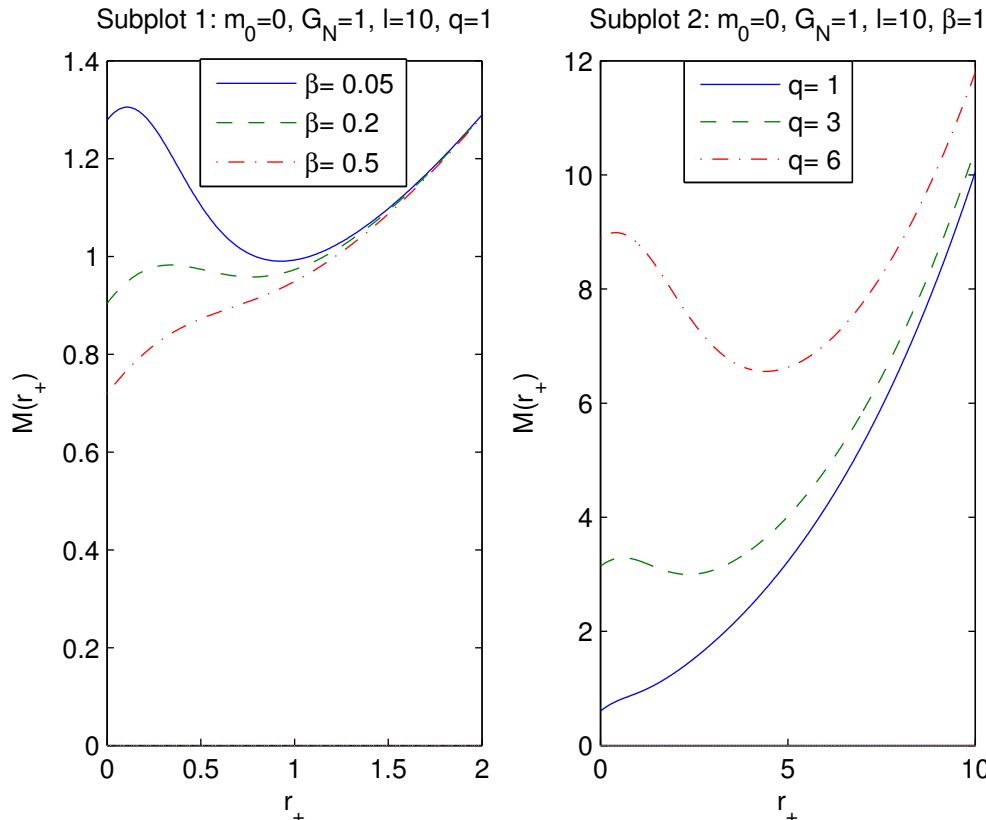

**Figure 2.** The function $M(r_+)$ at $m_0 = 0$, $G_N = 1$. According to Figure 2, left panel, the black hole mass $M(r_+)$ decreases, at fixed $r_+$ and $q$, when the coupling $\beta$ increases. In accordance with right panel, when magnetic charge $q$ increases, at fixed $r_+$ and $\beta$, the event horizon radius also increases.

With the help of Equation (17) we find

$$dM(r_+) = \left[\frac{1}{2} + \frac{3r_+^2}{2l^2} - \frac{q^2 r_+}{2(r_+^3 + (\beta q^2)^{3/4})}\right] dr_+ - \frac{r_+^3}{l^3} dl$$

$$+ \left[-\frac{\sqrt{q}\, g(r_+)}{8\beta^{1/4}} + \frac{\sqrt{3q}\,\pi}{8\beta^{1/4}} + \frac{q r_+^2}{4[r_+^3 + (\beta q^2)^{3/4}]}\right] dq$$

$$+ \left[\frac{q^{3/2} g(r_+)}{48\beta^{5/4}} - \frac{q^{3/2}\pi}{16\sqrt{3}\beta^{5/4}} + \frac{q^2 r_+^2}{8\beta[r_+^3 + (\beta q^2)^{3/4}]}\right] d\beta. \tag{18}$$

The Hawking temperature is given by

$$T = \frac{f'(r)|_{r=r_+}}{4\pi}, \tag{19}$$

where $f'(r) = \partial f(r)/\partial r$. By virtue of Equations (12) and (19), one obtains the Hawking temperature

$$T = \frac{1}{4\pi}\left[\frac{1}{r_+} + \frac{3r_+}{l^2} - \frac{q^2}{r_+^3 + (\beta q^2)^{3/4}}\right]. \tag{20}$$

Equation (20) is converted into the Hawking temperature of Maxwell-AdS black hole as $\beta \to 0$. Making use of Equations (15), (18), and (20), we find the first law of black hole thermodynamics

$$dM = TdS + VdP + \Phi dq + \mathcal{B}d\beta. \tag{21}$$

Comparing Equation (18) with (21), we obtain the magnetic potential $\Phi$ and the vacuum polarization $\mathcal{B}$

$$\Phi = -\frac{\sqrt{q}g(r_+)}{8\beta^{1/4}} + \frac{\sqrt{3q}\pi}{8\beta^{1/4}} + \frac{qr_+^2}{4[r_+^3 + (\beta q^2)^{3/4}]},$$

$$\mathcal{B} = \frac{q^{3/2}g(r_+)}{48\beta^{5/4}} - \frac{q^{3/2}\pi}{16\sqrt{3}\beta^{5/4}} + \frac{q^2 r_+^2}{8\beta[r_+^3 + (\beta q^2)^{3/4}]}. \tag{22}$$

The plots of $\Phi$ and $\mathcal{B}$ vs. $r_+$ are depicted in Figure 2.

According to Figure 3 (subplot 1), when parameter $\beta$ increases the magnetic potential $\Phi$ decreases. As $r_+ \to \infty$ the magnetic potential vanishes ($\Phi(\infty) = 0$), but at $r_+ = 0$ $\Phi$ is finite. Figure 3 (subplot 2) shows that at $r_+ = 0$ the vacuum polarization is finite and when $r_+ \to \infty$, $\mathcal{B}$ is zero ($\mathcal{B}(\infty) = 0$).

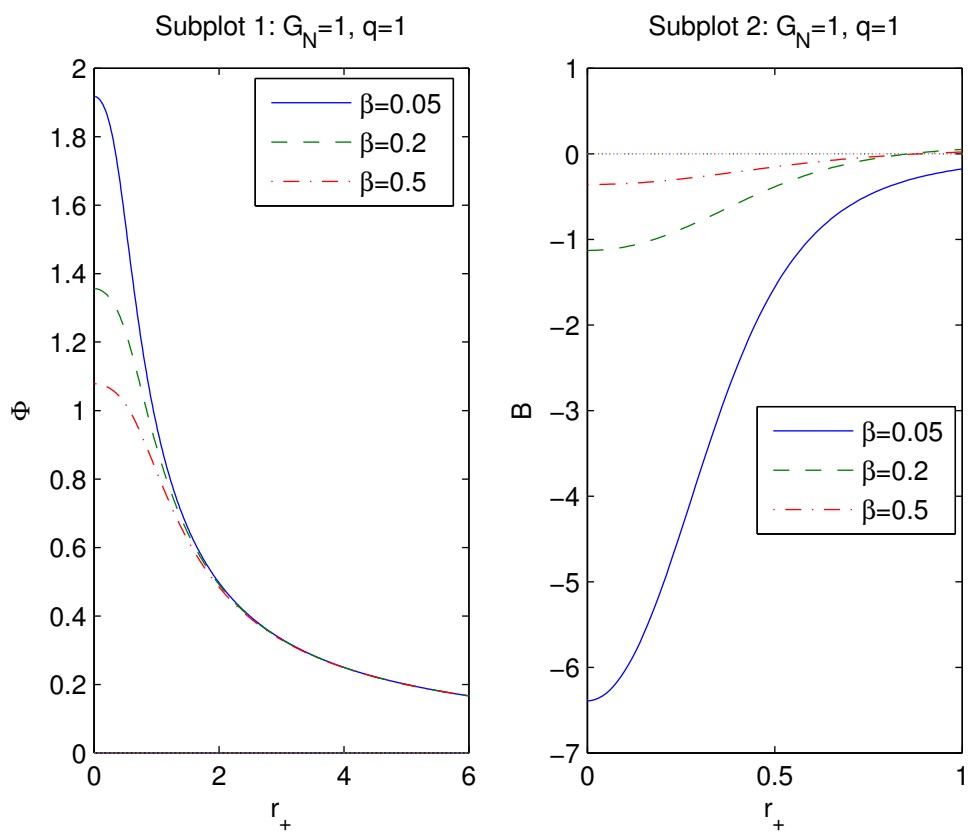

**Figure 3.** The functions $\Phi$ and $\mathcal{B}$ vs. $r_+$ at $q = 1$. The solid curve in subplot 1 is for $\beta = 0.05$, the dashed curve is for $\beta = 0.2$, and the dashed-dotted curve is for $\beta = 0.5$. It follows that the magnetic potential $\Phi$ is finite at $r_+ = 0$ and becomes zero at $r_+ \to \infty$. If coupling $\beta$ increases the magnetic potential decreases. The function $\mathcal{B}$ in subplot 2 vanishes as $r_+ \to \infty$ and is finite at $r_+ = 0$.

Making use of Equations (15), (16), and (22), one can verify that the generalized Smarr relation

$$M = 2ST - 2PV + q\Phi + 2\beta\mathcal{B} \tag{23}$$

holds.

## 4. Thermodynamics of Black Hole

With the help of Equation (20), one finds the black hole equation of state

$$P = \frac{T}{2r_+} - \frac{1}{8\pi r_+^2} + \frac{q^2}{8\pi r_+[r_+^3 + (\beta q^2)^{3/4}]}. \tag{24}$$

Equation (24), as $\beta \to 0$, is converted into charged Maxwell-AdS black hole equation of state [45]. Equation (24) is similar to the Van der Waals equation of state if the specific volume reads $v = 2l_P r_+$ ($l_P = \sqrt{G_N} = 1$) [45]. Following that, Equation (24) becomes

$$P = \frac{T}{v} - \frac{1}{2\pi v^2} + \frac{2q^2}{\pi v[v^3 + 8(\beta q^2)^{3/4}]}. \tag{25}$$

The inflection points in the $P - v$ diagrams (critical points) may be obtained by equations

$$\frac{\partial P}{\partial v} = -\frac{T}{v^2} + \frac{1}{\pi v^3} + \frac{8q^2(-v^6 + (q\sqrt{\beta})^3)}{\pi v^2[v^3 + 8(\beta q^2)^{3/4}]^2} = 0,$$

$$\frac{\partial^2 P}{\partial v^2} = \frac{2T}{v^3} - \frac{3}{\pi v^4} + \frac{8q^2[5v^6 + 8(\beta q^2)^{3/4}v^3 + 32(\beta q^2)^{3/2}]}{\pi v^3[v^3 + 8(\beta q^2)^{3/4}]^3} = 0. \tag{26}$$

By virtue of Equation (26), one finds the critical points equation

$$[v_c^3 + 8(\beta q^2)^{3/4}]^3 - 24q^2 v_c^4[v_c^3 - 4(\beta q^2)^{3/4}] = 0. \tag{27}$$

Making use of Equation (26), we obtain the critical temperature and pressure

$$T_c = \frac{1}{\pi v_c} - \frac{8q^2[v^3 + 2(\beta q^2)^{3/4}]}{\pi[v^3 + 8(\beta q^2)^{3/4}]^2}, \tag{28}$$

$$P_c = \frac{1}{2\pi v_c^2} - \frac{6q^2 v_c^2}{\pi(v_c^3 + 8(\beta q^2)^{3/4})^2}. \tag{29}$$

The solutions (approximate) $v_c$ to Equation (27), critical temperatures $T_c$, and pressures $P_c$ are presented in Table 1.

**Table 1.** Critical values of the specific volume, temperatures, and pressures at $q = 1$.

| $\beta$ | 0.1 | 0.2 | 0.4 | 0.5 | 0.7 | 0.8 | 0.9 | 1 |
|---|---|---|---|---|---|---|---|---|
| $v_c$ | 4.790 | 4.708 | 4.552 | 4.472 | 4.297 | 4.196 | 4.080 | 3.936 |
| $T_c$ | 0.0438 | 0.0442 | 0.0448 | 0.0452 | 0.0459 | 0.0463 | 0.0467 | 0.0472 |
| $P_c$ | 0.0034 | 0.0035 | 0.0036 | 0.0037 | 0.0038 | 0.0039 | 0.0040 | 0.0041 |

The $P - v$ diagrams are given in Figure 4.

At $q = 1$, $\beta = 0.5$ the critical specific volume is $v_c \approx 4.472$ and the critical temperature is $T_c = 0.0452$. Figure 4 shows that at some point the pressure is zero corresponding to the black hole remnant. Following that, if the specific volume increases the pressure increases and the pressure has a maximum. Then the pressure decreases, which is similar to ideal gas. At the critical values we have similarities, with Van der Waals liquid behavior having the inflection point. Making use of Equations (27)–(29) and for small $\beta$ one finds

$$v_c^2 = 24q^2 + \mathcal{O}(\beta), \quad T_c = \frac{1}{3\sqrt{6}\pi q} + \mathcal{O}(\beta), \quad P_c = \frac{1}{96\pi q^2} + \mathcal{O}(\beta). \tag{30}$$

At $\beta = 0$ in Equation (30), we obtain the critical points of charged AdS black hole [25]. Then the critical ratio becomes

$$\rho_c = \frac{P_c v_c}{T_c} = \frac{3}{8} + \mathcal{O}(\beta), \tag{31}$$

with the value $\rho_c = 3/8$ corresponding to the Van der Waals fluid.

The Gibbs free energy for fixed charge $q$, coupling $\beta$, and pressure $P$ is given by

$$G = M - TS, \tag{32}$$

where $M$ is considered a chemical enthalpy. Making use of Equations (16), (17), (20), and (32), we obtain

$$G = \frac{r_+}{4} - \frac{2\pi r_+^3 P}{3} + \frac{\pi q^{3/2}}{4\sqrt{3}\beta^{1/4}} + \frac{q^2 r_+^2}{4[r_+^3 + (\beta q^2)^{3/4}]} - \frac{q^{3/2}g(r_+)}{12\beta^{1/4}}. \tag{33}$$

The plot of the Gibbs free energy $G$ versus $T$ for $\beta = 0.5$ and $v_c \approx 4.472$, $T_c \approx 0.0452$ is depicted in Figure 5. We took into consideration that $r_+$ is the function of $P$ and $T$ (see Equation (24)).

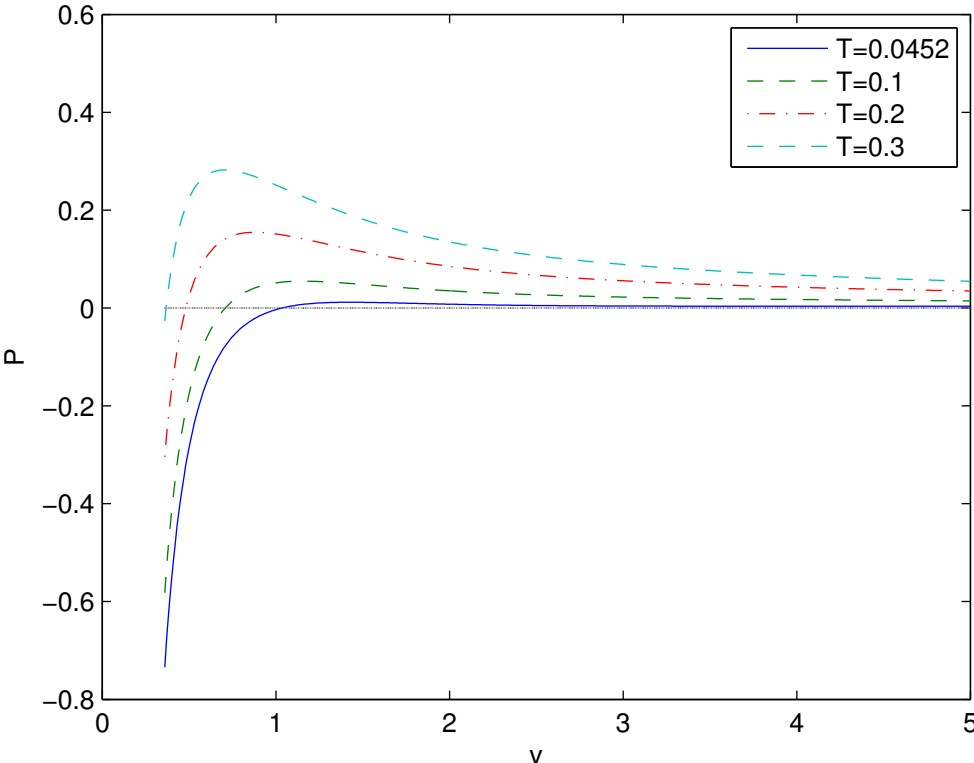

**Figure 4.** The function $P(v)$ at $q = 1$, $\beta = 0.5$. The critical isotherm corresponds to $T_c \approx 0.0452$ possessing the inflection point.

Subplots 1 and 2 at $P < P_c$ show first-order phase transitions (between small and large black holes for $T < T_c$) similar to liquid–gas transitions with the 'swallowtail' behavior. In accordance with subplot 3, the second order phase transition for $P = P_c$ takes place. Subplot 4 corresponds to the case $P > P_c$, where there are not phase transitions.

The entropy $S$ vs. temperature $T$ at $q = \beta = 1$ is given in Figure 6. Figure 6 (subplots 1 and 2) shows that entropy is ambiguous function of the temperature and, therefore, first-order phase transitions take place. According to subplot 3, the second-order phase

transition occurs. The critical point separates low and high entropy states. In accordance with subplot 4, there are not phase transitions at $q = \beta = 1$, $P = 0.005$.

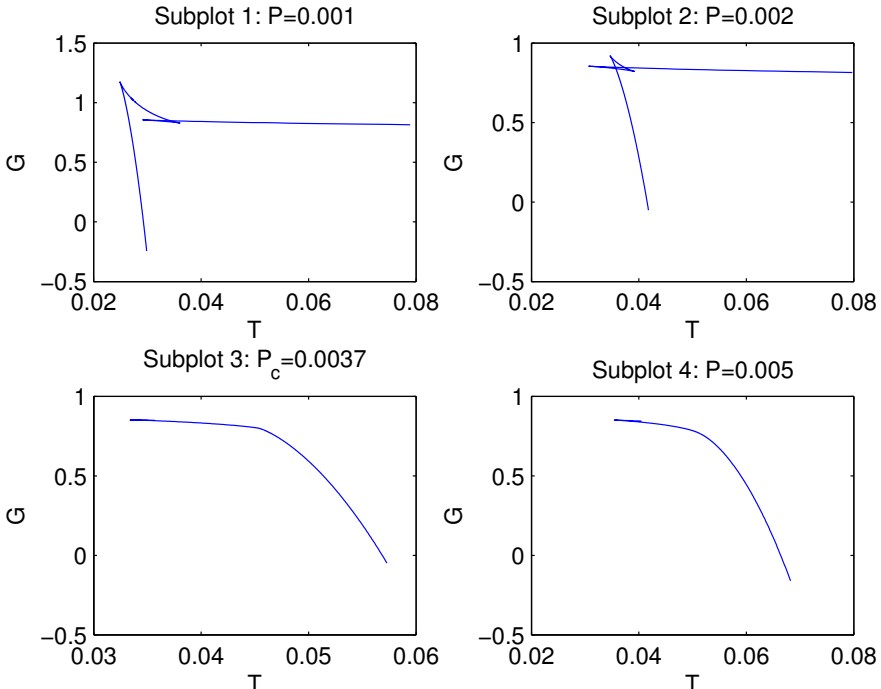

**Figure 5.** The plots of the Gibbs free energy $G$ vs. $T$ at $q = 1$, $\beta = 0.5$. According to subplots 1 and 2 we have the 'swallowtail' plots with first-order phase transitions. Subplot 3 shows the second-order phase transition with $P = P_c \approx 0.0037$. Subplot 4 shows the case $P > P_c$ with non-critical behavior of the Gibbs free energy.

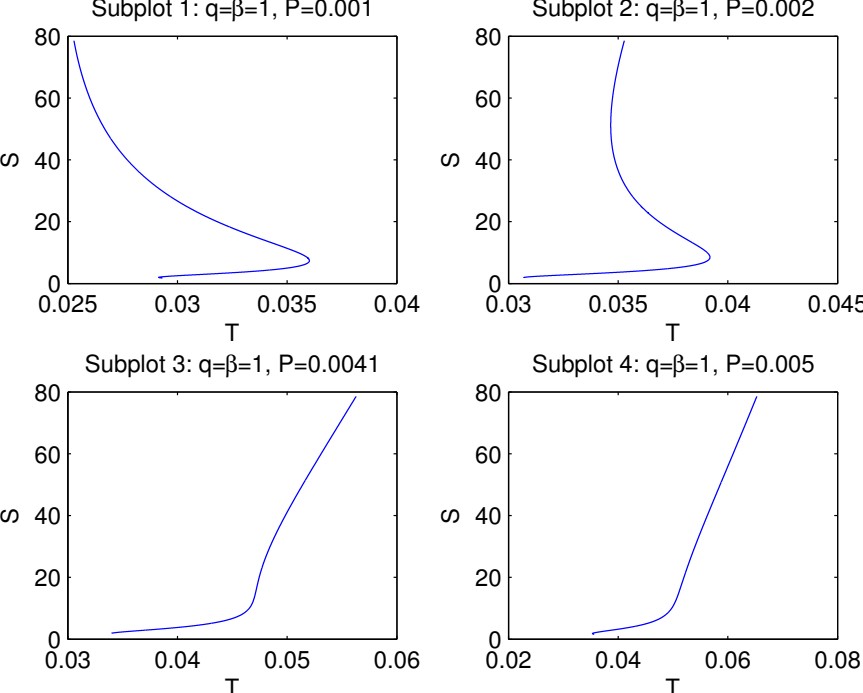

**Figure 6.** The plots of entropy $S$ vs. temperature $T$ at $q = 1$, $\beta = 0.5$. According to subplots 1 and 2 (in some range of $T$), entropy is an ambiguous function of the temperature, and first-order phase transitions occur. In accordance with subplot 3, the second-order phase transition takes place.

Let us study the local stability of black holes by considering the heat capacity which is given by

$$C_q = T\left(\frac{\partial S}{\partial T}\right)_q = \frac{T\partial S/\partial r_+}{\partial T/\partial r_+} = \frac{2\pi r_+ T}{G_N \partial T/\partial r_+}. \tag{34}$$

Equation (34) shows that when the Hawking temperature has an extreme the heat capacity diverges, and the black hole phase transition occurs. The plot of the Hawking temperature is given in Figure 7 for parameters $\beta = 0.1, 0.3, 1$ ($l = q = 1$).

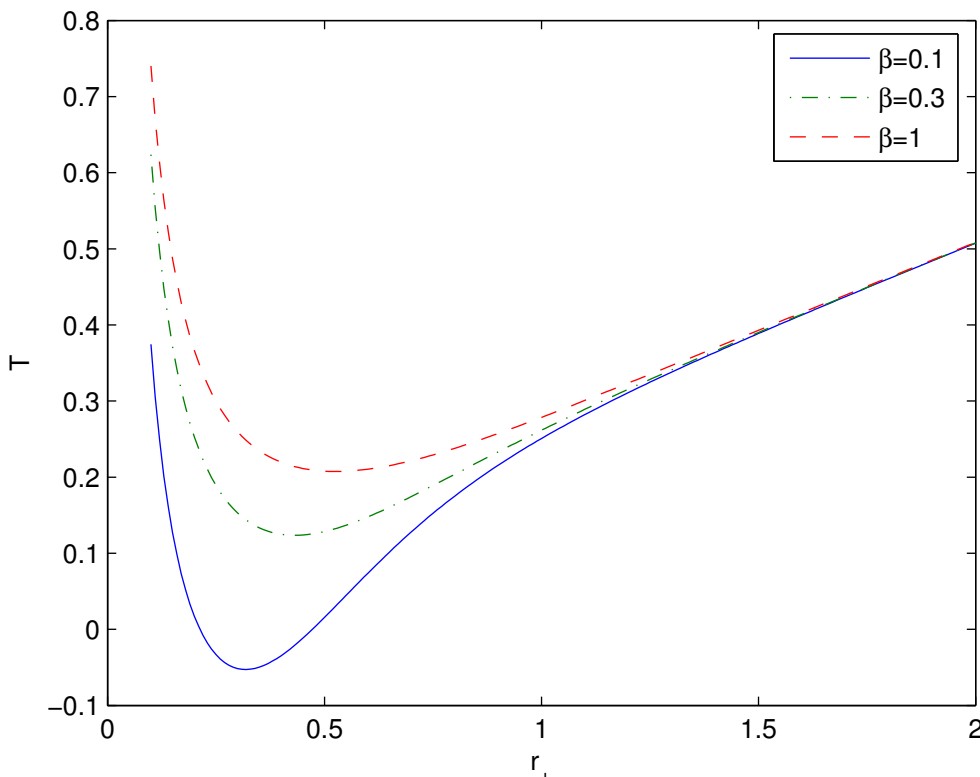

**Figure 7.** The plots of the Hawking temperature $T$ versus horizon radius $r_+$ at $l = q = 1$, $\beta = 0.1, 0.3, 1$. Figure shows that the Hawking temperature has a minimum.

In accordance with Figure 7, the Hawking temperature possesses a minimum and the heat capacity diverges. Figure 7 shows that there is a region where the Hawking temperature is negative and, therefore, in this interval of event horizon radiuses, black holes do not exist. For the case $l = q = 1$, $\beta = 0.1$, equation $T = 0$ has two real roots $r_1 \approx 0.213$ and $r_2 \approx 0.472$. The plot of the heat capacity (34) at $q = l = 1$, $\beta = 0.1$ ($G_N = 1$) is depicted in Figure 8. In accordance with Figure 8, the heat capacity has a singularity in the point where the Hawking temperature has a minimum. The heat capacity diverges ($\partial T/\partial r_+ = 0$) at $r_3 \approx 0.318$.

One can see from Equation (34) that when the Hawking temperature is a large black hole in the points where the heat capacity possesses a singularity. In the region where the heat capacity is positive the black hole is stable, otherwise the black hole is unstable. At $r_2 > r_+ > r_1$ the Hawking temperature is negative but at $r_+ > r_2$ the Hawking temperature and the heat capacity are positive and the black hole is stable.

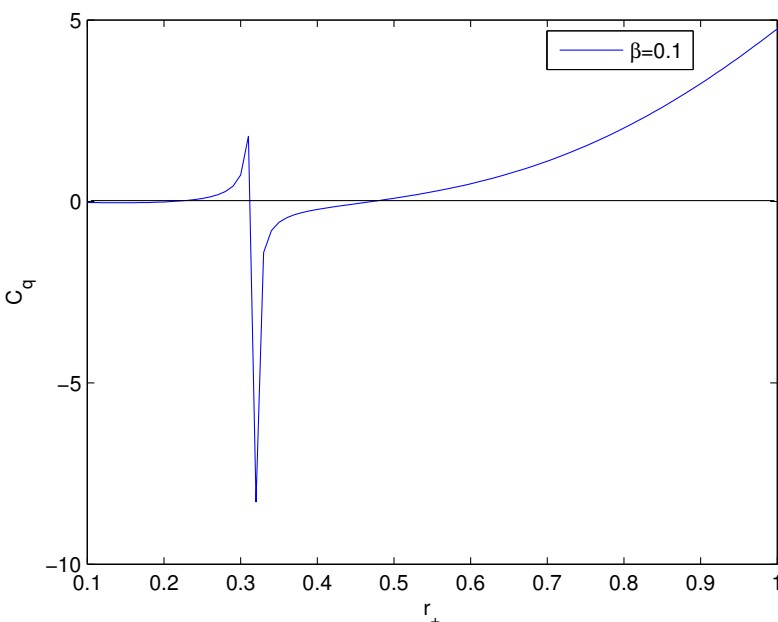

**Figure 8.** The plot of the heat capacity $C_q$ versus horizon radius $r_+$ at $l = q = 1$, $\beta = 0.1$. According to the figure, the heat capacity has a singularity where the Hawking temperature possesses a minimum.

*Reentrant Phase Transitions*

The reentrant phase transition was firstly observed in a nicotine/water mixture [48]. It was also discussed in higher dimensions [49] and in spinning Kerr-AdS black holes [50]. The phenomenon of reentrant phase transition is described in multi-component fluids [51]. The reentrant phase transition (zeroth-order phase transition) takes place when a system possesses a transition from one phase to another phase and then goes back to the first phase. In this process one thermodynamic variable is changed but others remain constant. As the pressure increases from 0.001 to 0.002 in Figure 8 (from panel 1 to panel 2), there will be a large-small-large reentrant phase transition. In our model there is the global minimum of the Gibbs free energy with a jump depicted in Figure 9 (for an example) for $P = 0.002$.

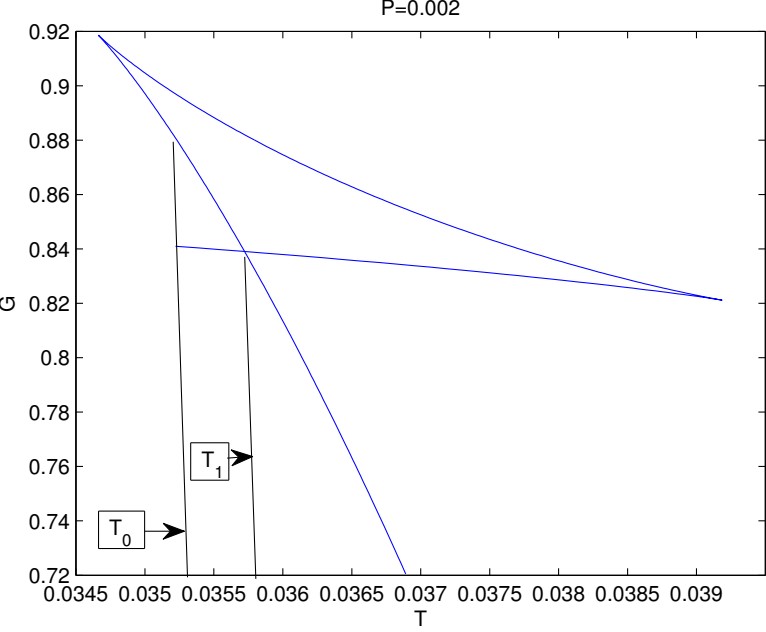

**Figure 9.** Reentrant phase transition. There is a finite jump in the Gibbs free energy showing the zeroth-order phase transition.

When $T$ decreases, the black hole follows the lower vertical curve until $T = T_1$. Then it coincides with the upper horizontal line corresponding to small stable black holes and undergoes a first-order large-small black hole phase transition. As $T$ decreases up to $T = T_0$, the Gibbs free energy $G$ possesses a discontinuity at its global minimum. When $T$ continues to decrease, the system goes to the stable black holes. Thus, the zeroth-order phase transition occurs between small and large black holes.

## 5. Joule–Thomson Expansion

During the Joule–Thomson isenthalpic expansion, the enthalpy (the mass $M$) is constant. The cooling–heating phases are described by the Joule–Thomson coefficient.

$$\mu_J = \left(\frac{\partial T}{\partial P}\right)_M = \frac{1}{C_P}\left[T\left(\frac{\partial V}{\partial T}\right)_P - V\right] = \frac{(\partial T/\partial r_+)_M}{(\partial P/\partial r_+)_M}. \tag{35}$$

Equation (35) shows that the Joule–Thomson coefficient is the slope of the $P - T$ function. At the inversion temperature $T_i$ the sign of $\mu_J$ is changed, and $T_i$ can be found by equation $\mu_J(T_i) = 0$. In the cooling phase ($\mu_J > 0$) initial temperature is higher than inversion temperature $T_i$ and the final temperature decreases. If the initial temperature is lower than $T_i$ then the final temperature increases in accordance with the heating phase ($\mu_J < 0$). Making use of Equation (35) and taking into account equation $\mu_J(T_i) = 0$, we obtain

$$T_i = V\left(\frac{\partial T}{\partial V}\right)_P = \frac{r_+}{3}\left(\frac{\partial T}{\partial r_+}\right)_P. \tag{36}$$

The inversion temperature separates cooling and heating processes. The inversion temperature line goes through $P - T$ diagrams maxima [31,32]. Equation (24) may be represented as equation of state

$$T = \frac{1}{4\pi r_+} + 2Pr_+ - \frac{q^2}{4\pi\left(r_+^3 + (\beta q^2)^{3/4}\right)}. \tag{37}$$

At $\beta = 0$ Equation (37) is converted into equation of state of Maxwell-AdS black holes. From Equation (17) and using equation $P = 3/(8\pi l^2)$, one obtains

$$P = \frac{3}{4\pi r_+^3}\left[M(r_+) - \frac{r_+}{2} - \frac{\pi q^{3/2}}{4\sqrt{3}\beta^{1/4}} + \frac{q^{3/2}g(r_+)}{12\beta^{1/4}}\right]. \tag{38}$$

We depicted the $P - T$ isenthalpic diagrams in Figure 9 by taking into account Equations (37) and (38). Figure 10 shows that the inversion $P_i - T_i$ diagram crosses maxima of isenthalpic curves.

Making use of Equations (24), (36), and (37), we find the inversion pressure $P_i$

$$P_i = \frac{3q^2\left(2r_+^3 + (\beta q^2)^{3/4}\right)}{16\pi r_+\left(r_+^3 + (\beta q^2)^{3/4}\right)^2} - \frac{1}{4\pi r_+^2}. \tag{39}$$

By virtue of Equations (37) and (39) one obtains the inversion temperature

$$T_i = \frac{q^2\left(4r_+^3 + (q^2\beta)^{3/4}\right)}{8\pi(r_+^3 + (q^2\beta)^{3/4})^2} - \frac{1}{4\pi r_+}. \tag{40}$$

Substituting $P_i = 0$ in Equation (39), we find the equation for the minimum of the event horizon radius $r_{min}$

$$3q^2 r_{min}(2r_{min}^3 + (\beta q^2)^{3/4}) - 4\left(r_{min}^3 + (\beta q^2)^{3/4}\right)^2 = 0. \tag{41}$$

From Equations (40) and (41) at $\beta = 0$, one obtains minimum of the inversion temperature corresponding to Maxwell-AdS magnetic black holes

$$T_i^{min} = \frac{1}{6\sqrt{6}\pi q}, \quad r_h^{min} = \frac{\sqrt{6}q}{2}. \tag{42}$$

Making use of Equations (30) and (42) at $\beta = 0$, we find the relation $T_i^{min} = T_c/2$, which corresponds to electrically charged AdS black holes [30]. With the help of Equations (39) and (40) we plotted $P_i - T_i$ diagrams in Figure 6. According to Figure 6, the inversion point increases when the black hole mass increases. The inversion diagrams $P_i - T_i$ are depicted in Figures 10 and 11.

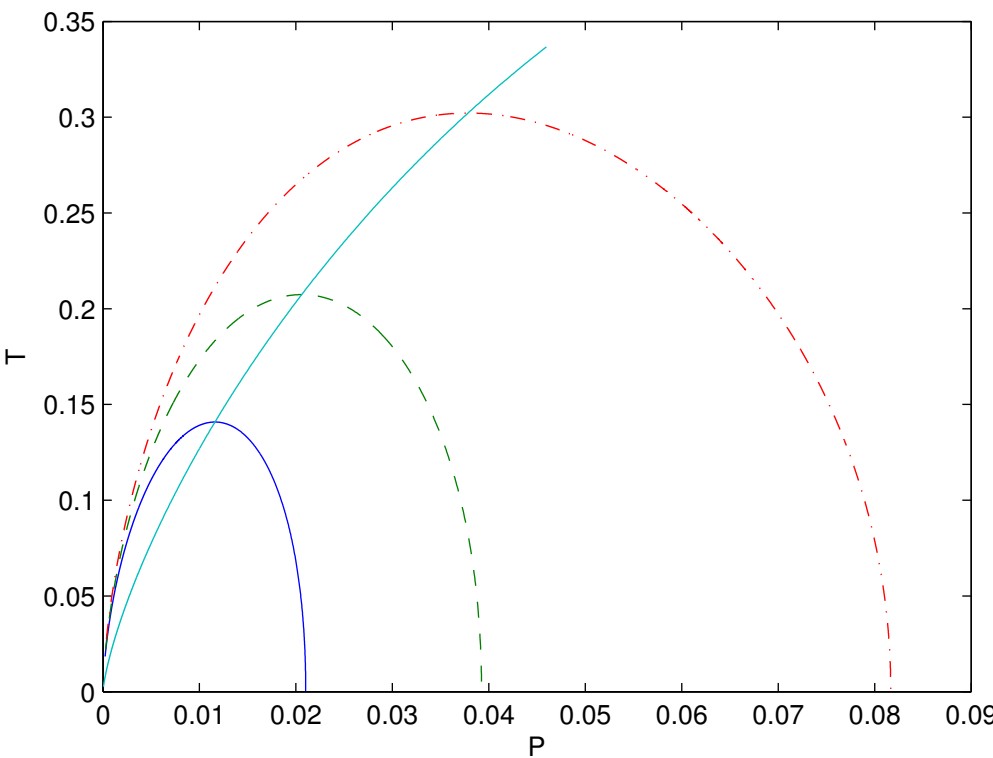

**Figure 10.** The plots of the temperature $T$ vs. pressure $P$ for $q = 30$, $\beta = 0.5$. The $P_i - T_i$ diagram goes via maxima of isenthalpic curves. The solid (blue) curve is for mass $M = 90$, the dashed (green) curve corresponds to $M = 100$, and the dashed–dotted (red) curve is for $M = 110$. The inversion temperature $T_i$ vs. pressure $P_i$ ($q = 30$, $\beta = 0.5$) is depicted by solid line. If black hole masses are increasing the inversion temperature, $T_i$ increases.

According to Figure 11, when magnetic charge $q$ increases then the inversion temperature increases. Figure 12 shows that when the coupling $\beta$ increases the inversion temperature decreases. From Equations (35), (37), and (38) we find

$$\left(\frac{\partial T}{\partial r_+}\right)_M = -\frac{1}{4\pi r_+^2} + 2P|_M + 2r_+\left(\frac{\partial P}{\partial r_+}\right)_M + \frac{3q^2 r_+^2}{4\pi[r_+^3 + (q^2\beta)^{3/4}]^2},$$

$$\left(\frac{\partial P}{\partial r_+}\right)_M = \frac{3}{4\pi r_+^4}\left[\frac{\sqrt{3}q^{3/2}\pi}{4\beta^{1/4}} - 3M + r_+ - \frac{q^{3/2}g(r_+)}{4\beta^{1/4}} + \frac{q^2 r_+^2}{2[r_+^3 + (\beta q^2)^{3/4}]}\right], \tag{43}$$

where $P|_M$ is given in Equation (38). Equations (35) and (43) define the Joule–Thomson coefficient as the function of the magnetic charge $q$, coupling $\beta$, black hole mass $M$, and

event horizon radius $r_+$. When the Joule–Thomson coefficient is positive ($\mu_J > 0$), a cooling process occurs. If $\mu_J < 0$, a heating process takes place.

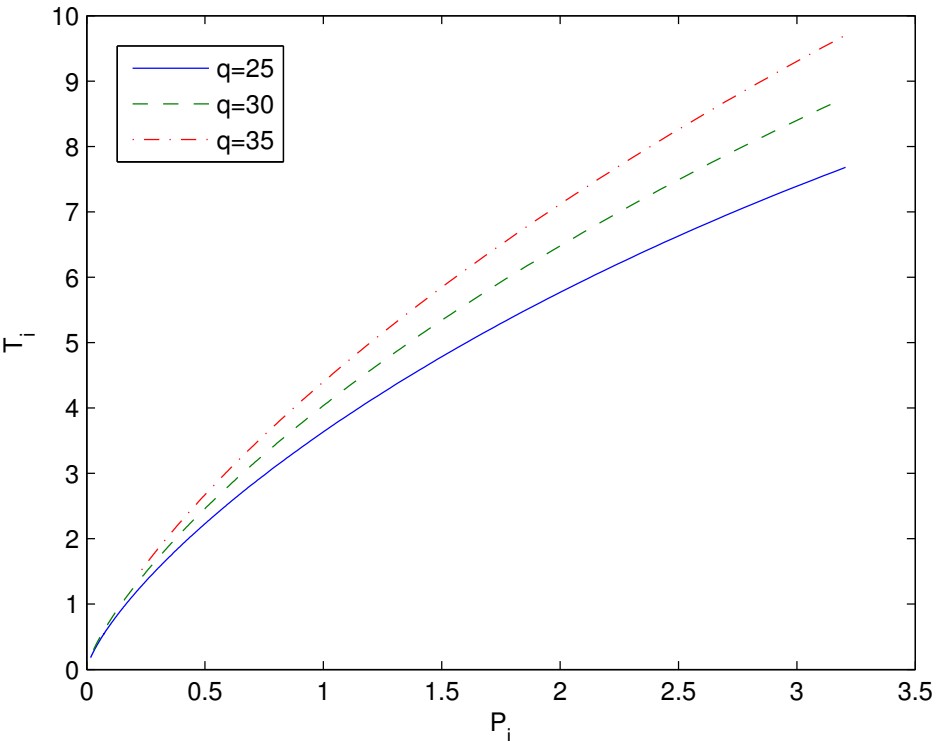

**Figure 11.** The inversion temperature $T_i$ vs. pressure $P_i$ at $q = 25$, 30, and 35, $\beta = 0.1$. When magnetic charge $q$ increases, the inversion temperature increases.

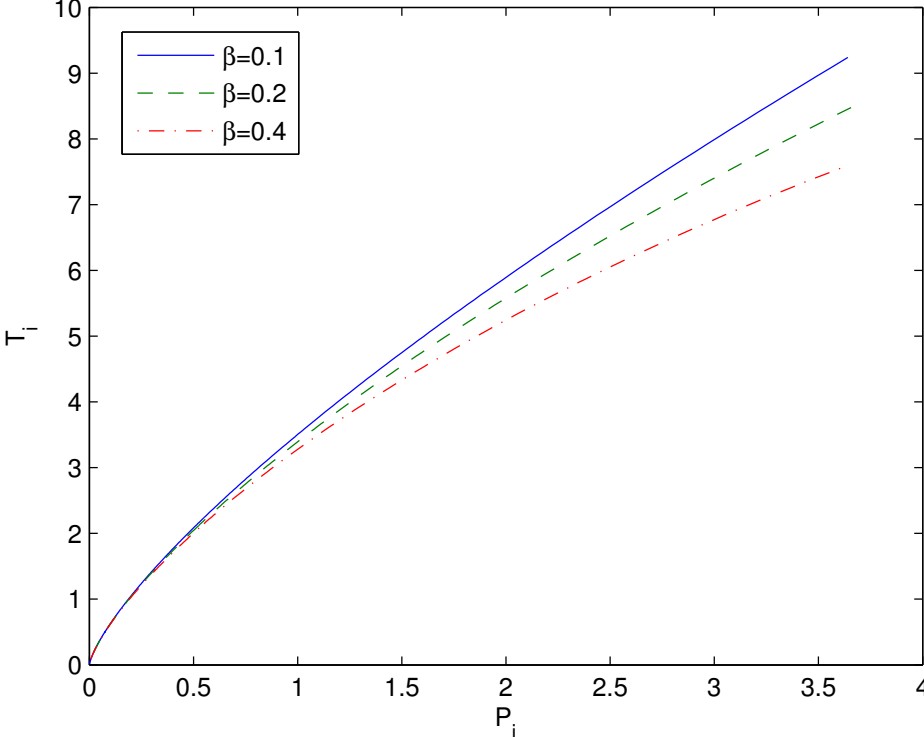

**Figure 12.** The inversion temperature $T_i$ vs. pressure $P_i$ at $\beta = 0.1$, 0.2 and 0.4, $q = 20$. Figure shows that if the coupling $\beta$ increases the inversion temperature decreases.

### 6. Summary

We obtained new magnetic black hole solution in Einstein-AdS gravity coupled to NED. It is shown that the principles of causality and unitarity occur for any magnetic induction fields. The metric and mass functions and corrections to the Reissner–Nordström solution were found. When coupling $\beta$ increases (at constant magnetic charge) the event horizon radius decreases. If magnetic charge increases (at constant coupling $\beta$) the event horizon radius increases. It was demonstrated that a spacetime singularity is present because the Kretschmann scalar is infinite at $r = 0$. The black holes thermodynamics in an extended phase space with negative cosmological constant (which is a thermodynamic pressure) was studied. In this approach, the mass of the black hole is the chemical enthalpy. The vacuum polarization, which is a thermodynamic quantity conjugated to coupling $\beta$, and thermodynamic potential, conjugated to magnetic charge, were obtained. We showed that the first law of black hole thermodynamics and the generalized Smarr formula take place. It was demonstrated that black hole thermodynamics is similar to the Van der Waals liquid–gas thermodynamics. We analyzed the Gibbs free energy and heat capacity showing phase transitions. We have analyzed zero-order, first-order, and second-order phase transitions. The critical ratio $\rho_c$ obtained is different from the Van der Waals value 3/8. We studied the black hole Joule–Thomson isenthalpic expansion and cooling and heating phase transitions. We found the inversion temperature which separates cooling and heating processes of black holes. There are similarities and differences in this and the past related papers. Expressions for the magnetic energy density, the mass and metric functions, the Hawking temperature, the magnetic potential, and the vacuum polarization, as well as the critical temperature and pressure, are different for models. It should be noted that a weak-gravity regime is released when $r$ goes to infinity. It follows from analytical expressions that as $r \to \infty$ a nonlinearity of NED disappears asymptotically.

**Funding:** This research received no external funding.

**Data Availability Statement:** Not applicable.

**Acknowledgments:** I wish to thank R. Mann for useful communications.

**Conflicts of Interest:** The author declares no conflict of interest.

### Appendix A

The Kretschmann scalar is defined as

$$K(r) \equiv R_{\mu\nu\alpha\beta}R^{\mu\nu\alpha\beta} = (f''(r))^2 + \left(\frac{2f'(r)}{r}\right)^2 + \left(\frac{2(f(r)-1)}{r^2}\right)^2. \tag{A1}$$

From Equation (12) (at $G_N = 1$) one finds

$$f'(r) = \frac{2m_0}{r^2} + \frac{q^{3/2}}{6r^2\sqrt[4]{\beta q^2}}\left(g(r) + \frac{\pi}{\sqrt{3}}\right) - \frac{q^{3/2}}{r^3 + \beta^{3/4}q^{3/2}} + \frac{2r}{l^2},$$

$$f''(r) = -\frac{4m_0}{r^3} - \frac{q^{3/2}}{3r^3\sqrt[4]{\beta q^2}}\left(g(r) + \frac{\pi}{\sqrt{3}}\right) + \frac{4r^3 + \beta^{3/4}q^{3/2}}{r(r^3 + \beta^{3/4}q^{3/2})^2)} + \frac{2}{l^2}, \tag{A2}$$

where

$$g(r) = \ln\frac{r^2 - \sqrt[4]{\beta q^2}r + \sqrt{\beta}q}{(r + \sqrt[4]{\beta q^2})^2} - 2\sqrt{3}\arctan\left(\frac{1 - 2r/\sqrt[4]{\beta q^2}}{\sqrt{3}}\right). \tag{A3}$$

Making use of Equations (12) and (A2) the Kretschmann scalar versus $r$ is plotted in Figure A1. As $r \to 0$ the Kretschmann scalar approaches to infinity showing a spacetime singularity at $r = 0$. But at small radiuses, close to the Planck length $l_P = \sqrt{G_N\hbar/c^3}$, one needs to take into account quantum effects [52]. The Kretschmann scalar becomes constant as $r \to \infty$.

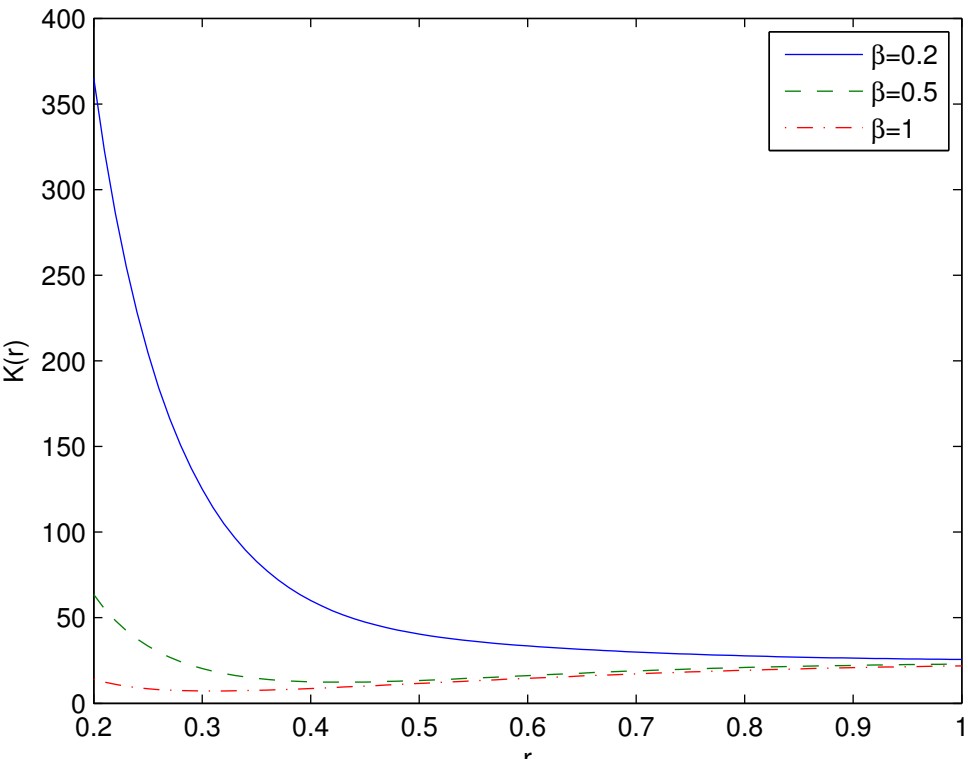

**Figure A1.** The plot of the function $K(r)$ vs. $r$ for $q = l = 1$, $m_0 = 0$. The solid line corresponds to $\beta = 0.2$, the dashed line corresponds to $\beta = 0.5$ and the dashed-dotted line corresponds to $\beta = 1$. The Kretschmann scalar approaches to infinity as $r \to 0$ showing a space-time singularity at $r = 0$. As $r \to \infty$ the Kretschmann scalar becomes constant.

According to Figure A1 the curvature invariant $K(r)$ is not bounded. Figure A1 shows that as the coupling $\beta$ increases (at fixed $q$ and $l$) the Kretschmann scalar decreases.

**Appendix B**

The NED models are viable if causality and unitarity principles take place. The causality principle requires that a group velocity of elementary excitations over a background field does not exceed the light speed in vacuum. The unitarity principle requires that the propagator residue has to be positive. These principles lead to requirements (in our notations) [53]

$$\mathcal{L}_{\mathcal{F}} \leq 0, \quad \mathcal{L}_{\mathcal{F}\mathcal{F}} \geq 0, \quad \mathcal{L}_{\mathcal{F}} + 2\mathcal{F}\mathcal{L}_{\mathcal{F}\mathcal{F}} \leq 0, \tag{A4}$$

were $\mathcal{L}_{\mathcal{F}} \equiv \partial\mathcal{L}/\partial\mathcal{F}$. Making use of of Equation (2) we obtain

$$\mathcal{L}_{\mathcal{F}} = -\frac{\beta\mathcal{F} + 2\sqrt[4]{2\beta\mathcal{F}}}{8\pi\sqrt[4]{2\beta\mathcal{F}}\left(1 + (2\beta\mathcal{F})^{3/4}\right)^2}, \quad \mathcal{L}_{\mathcal{F}\mathcal{F}} = \frac{9\beta}{32\pi\sqrt[4]{2\beta\mathcal{F}}\left(1 + (2\beta\mathcal{F})^{3/4}\right)^3},$$

$$\mathcal{L}_{\mathcal{F}} + 2\mathcal{F}\mathcal{L}_{\mathcal{F}\mathcal{F}} = -\frac{2(2\beta\mathcal{F})^{7/4} + 38\beta\mathcal{F} + 8\sqrt[4]{2\beta\mathcal{F}}}{32\pi\sqrt[4]{2\beta\mathcal{F}}\left(1 + (2\beta\mathcal{F})^{3/4}\right)^3}. \tag{A5}$$

Equation (A4) is satisfied for $\beta > 0$ and $\mathcal{F} = B^2/2 > 0$, i.e., for pure magnetic field. Thus, the principles of causality and unitarity occur for any magnetic induction fields.

## Note

[1]　We insert the factor $4\pi$ in the denominator of Equation (2) to use the Gaussian units compared to Heaviside–Lorentz units explored in [41].

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
