# Peer review of "Einstein-AdS Gravity Coupled to Nonlinear Electrodynamics, Magnetic Black Holes, Thermodynamics in an Extended Phase Space and Joule–Thomson Expansion"

_universe, doi:10.3390/universe9100456_

Round 1
Reviewer 1 Report
In the manuscript “Einstein-AdS Gravity coupled to nonlinear electrodynamics, magnetic black holes, thermodynamics in an extended phase space and Joule-Thomson expansion” the author addresses the physical implication of adopting a particular proposal for a nonlinear electrodynamics lagrangian that will be the “matter” lagrangian. This work is a continuation of a sequel of investigations of the author on this subject.
In this work the author shows that a generalising Smarr relation holds for the black holes derived form the spherically symmetric solution with the NED lagrangian under consideration. He further analyses te thermodynamical description of the black holes, finding an analogy with a Van der Waals fluid. He proceeds to address first and second-order phase transitions and the Joule-Thomson adiabatic expansion.
The work is well written and the results are sound and interesting.
Therefore I do recommend publication of the present manuscript in the journal Universe.
The quality of the english is fairly good. At times some article or verb is missing, e.g., at page 2, line 8 from the top: "... because of possibility..." should be "... because of the possibility...", or at page 5, line 3 from the top "singularity presents..." should rather be "singularity is present..."
So the author is advised to careful trimming of these type of issues (not a lot, in any case...).
Author Response
Thank you. I made corrections in English.
Reviewer 2 Report
In this paper, the Einstein-AdS gravity coupled with nonlinear electrodynamics is studied, and magnetic black holes and thermodynamics are explored in an extended phase space by using Joule-Thomson expansion. In particular, the metric and mass functions and corrections to the Reissner-Nordstrom solution are found, and the point that the black hole solutions can have one or two horizons is shown.
The arguments and descriptions are given in detail. The mathematical results are presented enough. If the following points are explained, this work will be suitable for publication.
1.Black hole solutions in general relativity with the negative cosmological constant (under the assumption of nonlinear electromagnetism) would have been studied in the past literature. The novel points found in this paper should be explained more clearly by comparing with the past related papers.
2.Thermodynamics and phase transitions of magnetically charged black holes in Anti-de Sitter spacetime are examined. The first law of black hole thermodynamics is formulated and the generalized Smarr relation is verified. From these analyses, what implications in terms of the adequacy of the introduction of such a nonlinear electromagnetism given in Eq. (2)?
3.The stability of the black hole and the (zeroth-, first- and second-order) phase transitions are analyzed by analyzing the Gibbs free energy and heat capacity with the Joule-Thomson expansion to examine the cooling and heating phase transitions. It is stated that the principles of causality and unitarity are satisfied in the present model. What physical mechanisms can lead to these consequences? (Apparently, the key point seems to be the specific introduction of the non-linearity of electromagnetism. Is such a nonlinear electromagnetism recovered in a weak-gravity regime asymptotically?)
Minor editing of English language required
Author Response
1.Black hole solutions in general relativity with the negative
cosmological constant (under the assumption of nonlinear
electromagnetism) would have been studied in the past literature. The
novel points found in this paper should be explained more clearly by
comparing with the past related papers.
A1) It was added in Summary: There are similarities and differences in
this and the past related papers. Expressions for the magnetic energy
density, the mass and metric functions, the Hawking temperature, the
magnetic potential and the vacuum polarization as well as the critical
temperature and pressure are different for models.
2.Thermodynamics and phase transitions of magnetically charged black
holes in Anti-de Sitter spacetime are examined. The first law of black
hole thermodynamics is formulated and the generalized Smarr relation is
verified. From these analyses, what implications in terms of the
adequacy of the introduction of such a nonlinear electromagnetism given
in Eq. (2)?
A2) We added in Introduction: It is worth mentioning that Lagrangians of
NED models in the weak-field limit are different. This leads to different
indexes of diffraction and birefringent effects.The similarities in the
behavior of critical isotherms, the magnetic potential, vacuum
polarization, the Gibbs free energy, and heat capacity take place for
Einstein-AdS gravity coupled to NED models.
3.The stability of the black hole and the (zeroth-, first- and second-
order) phase transitions are analyzed by analyzing the Gibbs free energy
and heat capacity with the Joule-Thomson expansion to examine the
cooling and heating phase transitions. It is stated that the principles
of causality and unitarity are satisfied in the present model. What
physical mechanisms can lead to these consequences? (Apparently, the key
point seems to be the specific introduction of the non-linearity of
electromagnetism. Is such a nonlinear electromagnetism recovered in a
weak-gravity regime asymptotically?)
A3) We added in Summary: It should be noted that a weak-gravity regime
is released when $r$ goes to infinity. It follows from analytical
expressions that as $r\rightarrow\infty$ a nonlinearity of NED disappears
asymptotically.
Reviewer 3 Report
In the article "Einstein-AdS gravity coupled to nonlinear electrodynamics, magnetic black holes, thermodynamics in an extended phase space and Joule–Thomson expansion" by S.I. Kruglov, the author explores the interaction between Einstein's gravity with a negative cosmological constant and nonlinear electrodynamics. The study involves deriving metric and mass functions along with corrections to the Reissner–Nordström solution, leading to black hole solutions with either one or two horizons. Thermodynamic properties and phase transitions of magnetically charged black holes in Anti-de Sitter space are investigated. The principles of causality and unitarity are affirmed within the model examined.
The article provides a detailed examination of Einstein-AdS gravity when coupled with nonlinear electrodynamics. The inclusion of the Joule–Thomson expansion and the analysis of phase transitions are notable. Overall, the article is a structured attempt to delve into complex theoretical concepts.
It would be beneficial to provide comments or insights regarding the results in de Sitter (dS) spacetime, as this could offer a more comprehensive understanding of the subject matter.
Author Response
In the article "Einstein-AdS gravity coupled to nonlinear electrodynamics
, magnetic black holes, thermodynamics in an extended phase space and
Joule–Thomson expansion" by S.I. Kruglov, the author explores the
interaction between Einstein's gravity with a negative cosmological
constant and nonlinear electrodynamics. The study involves deriving
metric and mass functions along with corrections to the Reissner–
Nordström solution, leading to black hole solutions with either one or
two horizons. Thermodynamic properties and phase transitions of
magnetically charged black holes in Anti-de Sitter space are investigated
. The principles of causality and unitarity are affirmed within the model
examined.
The article provides a detailed examination of Einstein-AdS gravity when
coupled with nonlinear electrodynamics. The inclusion of the Joule–
Thomson expansion and the analysis of phase transitions are notable.
Overall, the article is a structured attempt to delve into complex
theoretical concepts.
It would be beneficial to provide comments or insights regarding the
results in de Sitter (dS) spacetime, as this could offer a more
comprehensive understanding of the subject matter.
We added in Introduction:
Here, an attention is paid on gravity in the AdS (not in de Sitter)
spacetime because this case allows us to introduce a pressure which is
necessary to consider an extended phase space and thermodynamics. In
addition, only in this case the holographic principle occurs.